# Affine-Invariant Global Non-Asymptotic Convergence Analysis of BFGS under Self-Concordance

**Qiujiang Jin**
UT Austin
qiujiangjin0@gmail.com

**Aryan Mokhtari**
UT Austin & Google Research
mokhtari@austin.utexas.edu

## Abstract

In this paper, we establish global non-asymptotic convergence guarantees for the BFGS quasi-Newton method without requiring strong convexity or the Lipschitz continuity of the gradient or Hessian. Instead, we consider the setting where the objective function is strictly convex and strongly self-concordant. For an arbitrary initial point and any arbitrary positive-definite initial Hessian approximation, we prove global linear and superlinear convergence guarantees for BFGS when the step size is determined using a line search scheme satisfying the weak Wolfe conditions. Moreover, all our global guarantees are affine-invariant, with the convergence rates depending solely on the initial error and the strongly self-concordant constant. Our results extend the global non-asymptotic convergence theory of BFGS beyond traditional assumptions and, for the first time, establish affine-invariant convergence guarantees—aligning with the inherent affine invariance of the BFGS method.

## 1 Introduction

In this paper, we consider the convex optimization problem

$$\min_{x \in \mathbb{R}^d} f(x), \tag{1}$$

where the function $f$ is twice differentiable and *strictly* convex. We focus on quasi-Newton methods—iterative optimization algorithms that approximate the Hessian and its inverse using gradient information, making them efficient for large-scale problems where computing the Hessian is costly. Different variants update the Hessian approximation in distinct ways. The most famous quasi-Newton methods include the Davidon-Fletcher-Powell (DFP) method [1, 2], the Broyden-Fletcher-Goldfarb-Shanno (BFGS) method [3–6], the Symmetric Rank-One (SR1) method [7, 8], and the Broyden method [9]. There are also variants of these methods, including limited memory BFGS [10, 11], randomized quasi-Newton methods [12–16], and greedy quasi-Newton methods [15–18].

In this paper, we focus exclusively on the BFGS method, one of the most widely used and well-regarded quasi-Newton algorithms. Specifically, we analyze its convergence guarantees in the setting where the objective function is strictly convex and self-concordant and establish non-asymptotic guarantees for this case. Before highlighting our contributions, we first provide a summary of the existing convergence guarantees for BFGS as established in prior work.

**Classic asymptotic guarantees.** The local asymptotic superlinear convergence of quasi-Newton methods, including BFGS, has been established in several works [19–28]. Similarly, their global convergence under globalization strategies like line search and trust-region methods has been analyzed [8, 29–34]. However, these results are asymptotic and lack showing explicit rates.

**Non-asymptotic guarantees under stronger assumptions.** Recently, there were several breakthroughs regarding the non-asymptotic local superlinear convergence analysis of BFGS including [35–38] for the case that the objective function is strongly convex. More precisely, these works established

39th Conference on Neural Information Processing Systems (NeurIPS 2025).

an explicit superlinear rate of $\mathcal{O}(1/\sqrt{t})^t$ under the assumptions of strong convexity and Lipschitz continuity of the gradient and Hessian, given that the initial point is within a local neighborhood of the optimum and the initial Hessian approximation satisfies certain conditions. Later, these local analyses were extended and non-asymptotic global convergence rates of BFGS were established in [39–42] under similar assumptions on the objective function. In particular, [41] established global explicit superlinear convergence guarantees of the whole convex class of Broyden's family of quasi-Newton methods including both BFGS and DFP with step size satisfying the exact line search schemes. In a follow up work [42], the explicit global convergence rates for BFGS was established when deployed with an inexact line search satisfying the Armijo-Wolfe conditions. Specifically, these works show that when the objective is $\mu$-strongly convex, its gradient is $L$-Lipschitz smooth, and its Hessian is $K$-Lipschitz continuous, a global linear convergence rate of $(1 - 1/\kappa)^t$ can be achieved—matching that of gradient descent, where $\kappa = L/\mu$ is the condition number. Moreover, global superlinear convergence rates of $((d\kappa + C_0\kappa)/t)^t$ and $((C_0 d \log \kappa + C_0\kappa)/t)^t$ were established under specific choices of the initial Hessian approximation, where $d$ is the problem dimension, and $C_0$ is the initial function value gap between the initial iterate $x_0$ and the unique optimal solution $x_*$.

While these results represent significant progress in studying quasi-Newton methods, the established non-asymptotic guarantees for BFGS, and most quasi-Newton methods in general, have two major limitations. First, these results rely on relatively strong assumptions that may not hold in many practical settings. For instance, in the case of logistic regression, the loss is strictly convex but not necessarily strongly convex. Similarly, a log-barrier function does not satisfy the global Lipschitz condition for gradient. Second, all previously established non-asymptotic convergence rates for BFGS are not affine invariant, as they depend on parameters such as the strong convexity constant $\mu$, gradient Lipschitz constant $L$, and Hessian Lipschitz constant $K$, all of which vary under a change of basis or coordinate system in $\mathbb{R}^d$. In contrast, BFGS is affine invariant with respect to linear transformations of the variables. This means that the convergence behavior of BFGS remains unaffected by the choice of coordinate system and instead depends solely on the topological structure of $f$.

**Contributions:** We aim to address the discussed issues, and our main contributions are as follows:

- We establish global non-asymptotic linear and superlinear convergence rates for BFGS without requiring strong convexity or Lipschitz continuity of the gradient or Hessian. Instead, we consider functions that are strictly convex and strongly self-concordant. Our analysis provides explicit global convergence guarantees for BFGS when the step size is selected via a line search satisfying the weak Wolfe conditions. These guarantees hold for any initial point $x_0$ and any positive-definite initial Hessian approximation $B_0$.

- We derive explicit convergence rates for the BFGS method that are affine invariant. Specifically, our results show that both global linear and superlinear convergence rates depend solely on the strongly self-concordant constant, which remains invariant under linear transformations of the variables. To the best of our knowledge, these are the first theoretical convergence rates consistent with the affine invariance property of the BFGS method, reflecting its independence from the choice of coordinate system.

**Notation.** We denote the $l_2$-norm by $\|\cdot\|$ and the set of $d \times d$ symmetric positive definite matrices by $\mathbb{S}^d_{++}$. We write $A \preceq B$ if $B - A$ is positive semi-definite, and $A \prec B$ if it is positive definite. The trace and determinant of matrix $A$ are represented as $\mathbf{Tr}(A)$ and $\mathbf{Det}(A)$, respectively. For function $f$ that is strictly convex, we define the weighted norm $\|.\|_x$ as $\|u\|_x := \sqrt{u^\top \nabla^2 f(x) u}$

## 2 Background and Preliminaries

In this section, we provide a brief overview of the BFGS quasi-Newton method. At iteration $t$, $x_t$ denotes the current iterate, $g_t = \nabla f(x_t)$ the gradient of the objective function, and $B_t$ the Hessian approximation matrix. The general template of quasi-Newton methods update is given by

$$x_{t+1} = x_t + \eta_t d_t, \qquad d_t = -B_t^{-1} g_t, \tag{2}$$

where $\eta_t > 0$ is the step size. By defining the variable difference and the gradient difference as

$$s_t := x_{t+1} - x_t, \qquad y_t := \nabla f(x_{t+1}) - \nabla f(x_t), \tag{3}$$

we can present the Hessian approximation matrix update for BFGS as follows:

$$B_{t+1} := B_t - \frac{B_t s_t s_t^\top B_t}{s_t^\top B_t s_t} + \frac{y_t y_t^\top}{s_t^\top y_t}. \tag{4}$$

Further, if we define the inverse of Hessian approximation as $H_t := B_t^{-1}$, using the Sherman-Morrison-Woodbury formula, we have $H_{t+1} = (I - \frac{s_t y_t^\top}{y_t^\top s_t})H_t(I - \frac{y_t s_t^\top}{s_t^\top y_t}) + \frac{s_t s_t^\top}{y_t^\top s_t}$. Note that if the function $f$ is strictly convex – as considered in this paper – and the initial Hessian approximation matrix is positive definite, then $B_t \in \mathbb{S}_{++}^d$ for any iterations $t > 0$ (Chapter 6 [43]). In this paper, we focus on the analysis of BFGS when $\eta_t$ is selected based on the Armijo-Wolfe conditions, given by

$$f(x_t + \eta_t d_t) \leq f(x_t) + \alpha \eta_t \nabla f(x_t)^\top d_t, \tag{5}$$
$$\nabla f(x_t + \eta_t d_t)^\top d_t \geq \beta \nabla f(x_t)^\top d_t, \tag{6}$$

where $\alpha$ and $\beta$ are the line search parameters, satisfying $0 < \alpha < \beta < 1$ and $0 < \alpha < 1/2$.

**Affine Invariance property of BFGS.** From [44, 45], it is known that the iterates of BFGS are *affine invariant*. This property underscores the necessity of an analysis framework aligned with affine invariance, which is the main focus of our paper. We state the following proposition for completeness.

**Proposition 2.1.** *Let the iterations $\{x_t\}_{t=0}^{+\infty}$ be generated by the BFGS algorithm applied to the objective function $f(x)$, as defined in (2)-(4). Consider the iterates $\{\dot{x}_t\}_{t=0}^{+\infty}$ produced by applying BFGS to the transformed function $\phi(x) = f(Ax)$, where $A \in \mathbb{R}^{d \times d}$ is a non-singular matrix. Assume that the initializations satisfy $\dot{x}_0 = A^{-1}x_0$ and $\dot{B}_0 = A^\top B_0 A$. Then, for any $t \geq 0$, the following relationships hold: $\dot{x}_t = A^{-1}x_t$, $\dot{B}_t = A^\top B_t A$ and $\phi(\dot{x}_t) = f(x_t)$.*

## 2.1 Assumptions

Next, we state our assumptions and compare them with those used in prior work.

**Assumption 2.2.** The function $f$ satisfies the following conditions: (i) it is twice differentiable and strictly convex, and (ii) it is strongly self-concordant with parameter $M > 0$, i.e., for any $x, y, z \in \mathbb{R}^d$

$$\nabla^2 f(x) - \nabla^2 f(y) \preceq M\|x - y\|_z \nabla^2 f(y). \tag{7}$$

Our first assumption requires the objective function to be strictly convex, i.e., $\nabla^2 f(x) \succ 0$. This is indeed a weaker condition than the strong convexity assumptions used in prior works that establish non-asymptotic guarantees for BFGS, such as [35–42]. The second condition concerns strong self-concordance, which defines a subclass of self-concordant functions. Specifically, if $f$ is $M$-strongly self-concordant, then it is also $M/2$-self-concordant. To see this, fix $x \in \mathbb{R}^d$ and $u \in \mathbb{R}^d$. The inequality $u^\top(\nabla^2 f(x + tu) - \nabla^2 f(x))u \leq tM\|u\|_x^3$ holds, and dividing by $t$ and taking the limit as $t \to 0$ yields $D^3 f(x)[u, u, u] \leq M\|u\|_x^3$. A symmetric argument shows $|D^3 f(x)[u, u, u]| \leq M\|u\|_x^3$, implying that $f$ is self-concordant with parameter $M/2$. Moreover, Theorem 5.1.2 of [46] shows that the strong self-concordance parameter $M$ is affine invariant: for any non-singular $A \in \mathbb{R}^{d \times d}$, the function $\phi(x) = f(Ax)$ remains $M$-strongly self-concordant.

Next, we explain why our assumptions are strictly weaker than the more common conditions of strong convexity, Lipschitz gradient, and Lipschitz Hessian. Prior work (e.g., Example 4.1 in [17]) shows that if a function is strongly convex and its Hessian is Lipschitz with respect to a matrix $B \succeq 0$, then it is also strongly self-concordant. However, the converse does not hold: strong self-concordance does not imply strong convexity, gradient smoothness, or Lipschitz Hessian continuity.

As a concrete example, we can consider the log-sum-exp function formally defined as $f(x) = \log\left(\sum_{i=1}^n \exp(c_i^\top x - b_i)\right) + \sum_{i=1}^n (c_i^\top x)^2$, where $\{c_i\}_{i=1}^n \in \mathbb{R}^d$ and $\{b_i\}_{i=1}^n \in \mathbb{R}$. This function is not strongly convex with respect to the identity matrix $I$, due to the absence of explicit $\ell_2$ regularization. However, it can be shown to be strongly convex and have Lipschitz Hessian with respect to the matrix $B = \sum_{i=1}^n c_i c_i^\top$ (Note that this matrix could be possibly singular). As a result, it is strongly self-concordant but not strongly convex in the standard sense; check Appendix F. Other examples include the hard cubic function and the logistic regression objective discussed in Section 6. Another illustrative case is the log-barrier function $f(x) = -\log(1 - x^2)$, which is strongly self-concordant with $M = 4$ for $|x - y| \leq 1/2$, yet its gradient and Hessian are not Lipschitz continuous. Full detailed discussion for these examples is provided in Appendix F.

## 2.2 Definitions

Next, we state our definitions and notations. For any $A \in \mathbb{S}_{++}^d$, we define $\Psi(A)$ as

$$\Psi(A) := \mathbf{Tr}(A) - d - \log \mathbf{Det}(A). \tag{8}$$

This function characterizes the distance between matrix $A$ and the identity matrix $I$. Note that $\Psi(A) \geq 0$ for any $A \in \mathbb{S}_{++}^d$ and $\Psi(A) = 0$ if and only if $A = I$.

A common technique in the analysis of quasi-Newton methods involves the use of a reweighting matrix; see, e.g., [29]. We also use this approach in our analysis. Specifically, given any weight matrix $P \in \mathbb{S}_{++}^d$, we define the weighted versions of the vectors $g_t$, $s_t$, $y_t$, $d_t$ and the matrix $B_t$ as

$$\hat{g}_t := P^{-\frac{1}{2}} g_t, \quad \hat{s}_t := P^{\frac{1}{2}} s_t, \quad \hat{y}_t := P^{-\frac{1}{2}} y_t, \quad \hat{d}_t := P^{\frac{1}{2}} d_t, \quad \hat{B}_t := P^{-\frac{1}{2}} B_t P^{-\frac{1}{2}}. \tag{9}$$

The weight matrix $P$ plays fundamental role in our proof and the global linear and superlinear convergence rates are based on *different choices of* $P$. Note that the update rule for the weighted version of Hessian approximation matrices $\hat{B}_t$ is similar to the update rule of the unweighted $B_t$, i.e., $\hat{B}_{t+1} = \hat{B}_t - \frac{\hat{B}_t \hat{s}_t \hat{s}_t^\top \hat{B}_t}{\hat{s}_t^\top \hat{B}_t \hat{s}_t} + \frac{\hat{y}_t \hat{y}_t^\top}{\hat{s}_t^\top \hat{y}_t}$. We next introduce a common function in self-concordant analysis:

$$\omega(x) := x - \log(x + 1). \tag{10}$$

As shown in Lemma B.3, $\omega(x)$ is strictly increasing for $x > 0$. Hence, we can define its inverse function $\omega^{-1}(.)$ such that $\omega^{-1}(\omega(x)) = x$ for $x > 0$. It can be verified that $\omega^{-1}(x)$ is also strictly increasing for $x > 0$. Further, since $\omega(x)$ is a convex function, $\omega^{-1}(x)$ is concave. We use $\omega^{-1}$ to measure suboptimality of the iterates $\{x_t\}_{t=0}^{+\infty}$ and define the sequences $\{C_t\}_{t=0}^{+\infty}$ and $\{D_t\}_{t=0}^{+\infty}$ as

$$C_t := f(x_t) - f(x_*), \qquad D_t := 2\omega^{-1}\left(M^2 C_t / 4\right), \tag{11}$$

Indeed, both of the above sequences are always non-negative.

*Remark* 2.3. The expression $\omega^{-1}(.)$ frequently appears in our complexity bounds. To better understand this function and its approximation, as shown in Lemma B.3, we can use the approximation $\omega^{-1}(a) \approx (a + \sqrt{2a})$. Consequently, if $a < 1$, $\omega^{-1}(a) = \mathcal{O}(\sqrt{a})$, and if $a > 1$, $\omega^{-1}(a) = \mathcal{O}(a)$.

With these preliminaries, the next two sections prove global linear and superlinear convergence rates of BFGS for strictly convex, strongly self-concordant functions—rates that remain invariant under linear transformations, consistent with BFGS's affine invariance.

## 3 Global Linear Convergence Rates

In this section, we present the global linear convergence results of BFGS when the step size is selected based on the weak Wolfe conditions introduced in (5) and (6). Before we begin, we need to define the following weighted versions of the initial Hessian approximation matrix $B_0$:

$$\bar{B}_0 = \frac{\nabla^2 f(x_*)^{-\frac{1}{2}} B_0 \nabla^2 f(x_*)^{-\frac{1}{2}}}{1 + D_0}, \qquad \tilde{B}_0 = \nabla^2 f(x_*)^{-\frac{1}{2}} B_0 \nabla^2 f(x_*)^{-\frac{1}{2}}. \tag{12}$$

These two weighted versions of $B_0$ correspond to the weight matrices $P = (1 + D_0) \nabla^2 f(x_*)$ and $P = \nabla^2 f(x_*)$, respectively. They play a key role in the non-asymptotic analysis of BFGS for self-concordant functions. Next, we present our first global explicit linear convergence rate of BFGS for any initial point $x_0$ and any initial Hessian approximation matrix $B_0 \in \mathbb{S}_{++}^d$.

**Theorem 3.1.** *Suppose Assumption 2.2 holds. Let $\{x_t\}_{t\geq 0}$ be the iterates generated by BFGS, where the step size satisfies the Armijo-Wolfe conditions in (5) and (6). Recall $\Psi(\cdot)$ in (8), $D_0$ in (11) and $\bar{B}_0$ in (12). For any initial point $x_0 \in \mathbb{R}^d$ and any initial Hessian approximation $B_0 \in \mathbb{S}_{++}^d$, we have*

$$\frac{f(x_t) - f(x_*)}{f(x_0) - f(x_*)} \leq \left(1 - \frac{\alpha(1 - \beta)e^{-\frac{\Psi(\bar{B}_0)}{t}}}{(1 + D_0)^2}\right)^t. \tag{13}$$

*Moreover, when $t \geq \Psi(\bar{B}_0)$, we obtain that*

$$\frac{f(x_t) - f(x_*)}{f(x_0) - f(x_*)} \leq \left(1 - \frac{\alpha(1 - \beta)}{3(1 + D_0)^2}\right)^t. \tag{14}$$

Theorem 3.1 states that BFGS converges globally at a linear rate, influenced by the line search parameters (as expected), the term $\Psi(\bar{B}_0)$, which quantifies the discrepancy between the initial Hessian approximation and the optimal one, and $D_0$, which depends on the suboptimality of the initial function value and the strongly self-concordance parameter. To further simplify the expression, as shown in the second result, when $t \geq \Psi(\bar{B}_0)$, the linear convergence rate can be further simplified as $\mathcal{O}(1 - 1/(1 + D_0)^2)$. Hence, $D_0 = 2\omega^{-1}(M^2(f(x_0) - f(x_*)/4)$ indicates the rate.

Two remarks follow the above result. First, our global linear convergence rate does not require assuming strong convexity or gradient Lipschitz-ness. Second, the linear convergence rate is affine invariant across different linear systems, consistent with the affine invariance property of BFGS.

We emphasize that the proof of Theorem 3.1 for showing global linear convergence rate is fundamentally different from the analyses in prior work. Specifically, the results in [41, 38, 42] heavily depend on the strong convexity and gradient Lipschitz-ness assumptions to showcase a linear convergence rate: they use the Lipschitz continuity of the gradient to upper bound $\|y_t\|^2/s_t^\top y_t$ by $L$, and use $\mu$-strong convexity to establish the following lower bound $\|g_t\|^2/(f(x_t) - f(x_*)) \geq 2\mu$. These bounds are key to establishing the global linear rate of BFGS in prior work. In our setting such bounds do not hold and we do not have a universal upper bound on $\|y_t\|^2/s_t^\top y_t$ and a lower bound on $\|g_t\|^2/(f(x_t) - f(x_*))$. Instead, for the first bound, we transfer the inequality to the norm induced by the weight matrix $P = (1 + D_0)\nabla^2 f(x_*)$ and show under this norm and strong self-concordance assumption we have $\|\hat{y}_t\|^2/\hat{s}_t^\top \hat{y}_t \leq 1$. For the lower bound on $\|g_t\|^2/(f(x_t) - f(x_*))$, instead of a uniform lower bound, we show that it can be bounded below by $1/(1 + D_t)$, which is dependent on $x_t$, but we show that even this time-dependent lower bound is sufficient to establish a linear convergence rate for BFGS. For more details check the proofs of Lemma B.7 and Section C.2 in the Appendix.

The linear convergence result depends on $\Psi(\bar{B}_0)$, and hence the choice of $B_0$ affects the convergence rate. In practice, it is often a scaled identity and a common choice is $B_0 = cI$, where $c = (s^\top y)/\|s\|^2$, with $s = x_2 - x_1$, $y = \nabla f(x_2) - \nabla f(x_1)$, and $x_1, x_2$ as two randomly selected vectors. In the next corollary, we present our global linear rate when $B_0 = aI$ where $a > 0$ is an arbitrary constant.

**Corollary 3.2.** *Suppose Assumptions 2.2 holds, $\{x_t\}_{t\geq0}$ are generated by BFGS with step size satisfying the Armijo-Wolfe conditions in (5) and (6), and $x_0 \in \mathbb{R}^d$ is an arbitrary initial point. If the initial Hessian approximation matrix is set as $B_0 = aI$ for any $a > 0$, then we have that*

$$\frac{f(x_t) - f(x_*)}{f(x_0) - f(x_*)} \leq \left(1 - \frac{\alpha(1-\beta)e^{-\frac{\Delta_1}{t}}}{(1 + D_0)^2}\right)^t, \tag{15}$$

*where $\Delta_1 := \Psi(\frac{a\nabla^2 f(x_*)^{-1}}{1 + D_0})$ can be written as*

$$\Delta_1 = \mathbf{Tr}\left[\frac{a\nabla^2 f(x_*)^{-1}}{1 + D_0}\right] - d - \log \mathbf{Det}\left[\frac{a\nabla^2 f(x_*)^{-1}}{1 + D_0}\right]. \tag{16}$$

*Moreover, when $t \geq \Delta_1$, we obtain that*

$$\frac{f(x_t) - f(x_*)}{f(x_0) - f(x_*)} \leq \left(1 - \frac{\alpha(1-\beta)}{3(1 + D_0)^2}\right)^t. \tag{17}$$

Note that the proof of this corollary simply follows by setting $B_0 = aI$ in Theorem 3.1. The above result shows that by selecting $B_0 = aI$, the linear convergence rates of the BFGS method is totally determined by the initial suboptimality $D_0$ and the trace and determinant of the inverse matrix of the Hessian at $x_*$, which are also consistent with the affine invariance property of BFGS.

Next, we proceed to present an improved version of the result in Theorem 3.1, showing that after a sufficient number of iterations, the linear rate of BFGS becomes independent of $D_0$ and $B_0$.

**Theorem 3.3.** *Suppose Assumptions 2.2 holds, and let $\{x_t\}_{t\geq0}$ be the iterates generated by the BFGS method with the Armijo-Wolfe line search in (5) and (6). Recall the definition of $\Psi(\cdot)$ in (8), $D_0$ in (11) and $\bar{B}_0$, $\tilde{B}_0$ in (12). Then, for any initial point $x_0 \in \mathbb{R}^d$ and any initial Hessian approximation matrix $B_0 \in \mathbb{S}_{++}^d$, when $t \geq \Psi(\tilde{B}_0) + 3D_0(\Psi(\bar{B}_0) + \frac{3(1+D_0)^2}{\alpha(1-\beta)})$, we have*

$$\frac{f(x_t) - f(x_*)}{f(x_0) - f(x_*)} \leq \left(1 - \frac{2\alpha(1-\beta)}{3}\right)^t. \tag{18}$$

This theorem demonstrates that when the number of iterations is larger than $\Psi(\tilde{B}_0) + 3D_0(\Psi(\bar{B}_0) + \frac{3(1+D_0)^2}{\alpha(1-\beta)})$, BFGS with stepsize satisfying the Armijo-Wolfe conditions achieves an explicit linear convergence rate that is independent of the initial suboptimality $D_0$ and only determined by the line search parameters $\alpha$ and $\beta$ defined in (5) and (6). That said, the point that transition to this fast rate happens still depends on the choice of $x_0$ and $B_0$, as stated in Theorem 3.3. Similar to Corollary 3.2, next we present the special case of Theorem 3.3 of $B_0 = aI$ for any $a > 0$.

**Corollary 3.4.** *Suppose Assumptions 2.2 holds, $\{x_t\}_{t \geq 0}$ are generated by BFGS with step size satisfying the Armijo-Wolfe conditions in* (5) *and* (6)*, and $x_0 \in \mathbb{R}^d$ is an arbitrary initial point. If the initial Hessian approximation matrix is set as $B_0 = aI$ for any $a > 0$, then the following rate holds*

$$\frac{f(x_t) - f(x_*)}{f(x_0) - f(x_*)} \leq \left(1 - \frac{2\alpha(1-\beta)}{3}\right)^t, \tag{19}$$

*for all iterates satisfying $t \geq \Delta_2 + 3D_0\left(\Delta_1 + \frac{3(1+D_0)^2}{\alpha(1-\beta)}\right)$, where $\Delta_1$ is defined in* (16) *and*

$$\Delta_2 = \mathbf{Tr}(a\nabla^2 f(x_*)^{-1}) - d - \log \mathbf{Det}(a\nabla^2 f(x_*)^{-1}). \tag{20}$$

Note that both $\Delta_1$ and $\Delta_2$ are determined by the Hessian at the optimal solution $x_*$, while $\Delta_1$ also depends on the initial suboptimality error through $D_0$. In general, we do expect the convergence rates of BFGS to depend on the distance between $x_0$ and $x_*$, which is characterized by $D_0$ defined in (11) as well as the distance between the initial Hessian approximation matrix $B_0$ and the exact Hessian at optimal solution $x_*$, which is characterized by $\Delta_1$ and $\Delta_2$ when $B_0 = \alpha I$.

## 4 Global Superlinear Convergence Rates

Building on the established linear convergence results, we next establish our global superlinear convergence rate of BFGS. A key point in our analysis is that to reach the superlinear convergence stage, the unit step size must be chosen after some iterations. This is a necessary condition, as noted in several prior works [30–32, 29]. The fundamental methodology is to first establish the sufficient conditions of when the unit step size can be selected, i.e., when $\eta_t = 1$ satisfies the conditions in (5) and (6). Then, based on these conditions, we can prove that after some specific iterations $t_0$, the unit step size $\eta_t = 1$ is admissible for the inexact line search scheme except for a finite number of iterations, which leads to the final proof of the global non-asymptotic superlinear convergence rate.

Next, we proceed to establish under what conditions $\eta = 1$ is admissible. First, define $\rho_t$ as

$$\rho_t := \frac{-g_t^\top d_t}{\|\tilde{d}_t\|^2}, \quad \tilde{d}_t := \nabla^2 f(x_*)^{\frac{1}{2}} d_t, \quad \forall t \geq 0. \tag{21}$$

In the following lemma, we demonstrate that when $C_t = f(x_t) - f(x_*)$ is small enough and $\rho_t$ is close enough to 1, the unit step size $\eta_t = 1$ is admissible and meets the Armijo-Wolfe conditions.

**Lemma 4.1.** *Suppose Assumption 2.2 holds and define*

$$\delta_1 := \min\left\{\frac{1}{16}, \frac{4}{M^2}\omega\left(\frac{1}{32}\right), \frac{4}{M^2}\omega\left(\frac{\sqrt{2(1-\alpha)}-1}{2}\right), \frac{4}{M^2}\omega\left(\frac{1}{2}\left(\frac{1}{\sqrt{1-\beta}}-1\right)\right)\right\},$$

$$\delta_2 := \max\left\{\frac{15}{16}, \frac{1}{\sqrt{2(1-\alpha)}}\right\}, \quad \delta_3 := \frac{1}{\sqrt{1-\beta}}, \tag{22}$$

*which satisfy $0 < \delta_1 < \delta_2 < 1 < \delta_3$. If $C_t \leq \delta_1$ and $\delta_2 \leq \rho_t \leq \delta_3$, then $\eta_t = 1$ satisfies* (5) *and* (6)*.*

First, we highlight the key difference between Lemma 4.1 and prior results in [38, 42, 41]. The proof of Lemma 4.1 hinges on ensuring $f(x_t + d_t) \leq f(x_t)$, i.e., that a unit step yields a decrease in function value. Under Lipschitz continuity of the Hessian with constant $K$, the error of approximating $f(y)$ by its second-order Taylor expansion at $x$ is bounded by $\frac{K}{6}\|y - x\|^3$. Without this assumption, and under $M$-strongly self-concordant assumption, we instead use the bound $f(y) \leq f(x) + g(x)^\top(y - x) + \frac{4}{M^2}\omega_*\left(\frac{M}{2}\|y - x\|_x\right)$ for $\|y - x\|_x < \frac{2}{M}$, where $\omega_*(x) = -x - \log(1 - x)$ is defined for $x < 1$. As a result, the error is no longer cubic in $\|y - x\|$, making it more challenging to ensure a function

decrease. Nevertheless, we can still guarantee this property, with the main difference being that the error bound $\delta_1$ now depends on $\omega(x)$ defined above. See Lemma B.9 and Section C.4 for details.

The result in Lemma 4.1 shows that when $C_t \leq \delta_1$ and $\rho_t \in [\delta_2, \delta_3]$, we can choose the step size $\eta_t = 1$ at iteration $t$ of BFGS, as it satisfies the weak Wolfe conditions. Moreover, from the global non-asymptotic linear convergence rates of the last section, we can specify the $t_0$ such that for any $t \geq t_0$, the first condition $C_t \leq \delta_1$ always holds. Moreover, we can demonstrate that the second condition on $\rho_t$ is violated only for a finite number of iterations, i.e., the set of the indices that $\rho_t \notin [\delta_2, \delta_3]$ can be upper bounded by some constants. We formally present these results in the following lemma and the proofs are available in Appendix C.5.

**Lemma 4.2.** *Suppose Assumptions 2.2 holds and $\{x_t\}_{t\geq 0}$ are generated by BFGS with step size satisfying the Armijo-Wolfe conditions in (5)-(6). Recall the definition of $C_t$ in (11), $D_t$ in (11), $\Psi(\cdot)$ in (8), $\{\delta_i\}_{i=1}^3$ in (22), and $\bar{B}_0$, $\tilde{B}_0$ in (12). We have $C_t \leq \delta_1$ when $t \geq t_0$, where $t_0$ is defined as*

$$t_0 := \max \left\{ \Psi(\bar{B}_0), \ \frac{3(1+D_0)^2}{\alpha(1-\beta)} \log \frac{C_0}{\delta_1} \right\}. \tag{23}$$

*Moreover, the size of the set $I = \{t_0 \leq i \leq t-1 : \ \rho_t \notin [\delta_2, \delta_3]\}$ is at most*

$$|I| \leq \delta_4 \left( \Psi(\tilde{B}_0) + 2D_0 \left( \Psi(\bar{B}_0) + \frac{3(1+D_0)^2}{\alpha(1-\beta)} \right) \right), \quad \text{where} \ \delta_4 := \frac{1}{\min\{\omega(\delta_2 - 1), \omega(\delta_3 - 1)\}}. \tag{24}$$

The above lemma specifies the time instance $t_0$ for which $C_t \leq \delta_1$ is satisfied for any $t \geq t_0$ and for only a finite number of indices, the condition $\rho_t \in [\delta_2, \delta_3]$ does not hold. In practice, we always start with the unit step size when we implement the inexact line search scheme at iteration $t$ to check if $\eta_t = 1$ satisfies the Armijo-Wolfe conditions in (5) and (6). Hence, when $t \geq t_0$, only for a finite number of iterations that $\rho_t \notin [\delta_2, \delta_3]$, the unit step size is not selected. With all these points, we present the global superlinear convergence rate of BFGS for self-concordant functions.

**Theorem 4.3.** *Suppose Assumptions 2.2 holds and the iterates $\{x_t\}_{t\geq 0}$ are generated by BFGS with step size satisfying the Armijo-Wolfe conditions in (5) and (6). Recall the definition of $D_t$ in (11), $\Psi(\cdot)$ in (8), $\bar{B}_0$, $\tilde{B}_0$ in (12), and $\{\delta_i\}_{i=1}^4$ in (22), (24). Then, for any initial point $x_0 \in \mathbb{R}^d$ and any initial Hessian approximation matrix $B_0 \in \mathbb{S}_{++}^d$, the following global superlinear result holds:*

$$\frac{f(x_t) - f(x_*)}{f(x_0) - f(x_*)} \leq \left( \frac{\delta_6 t_0 + \delta_7 \Psi(\tilde{B}_0) + \delta_8 D_0 (\Psi(\bar{B}_0) + \frac{3(1+D_0)^2}{\alpha(1-\beta)})}{t} \right)^t,$$

*where $t_0$ is defined in (23), $\{\delta_i\}_{i=5}^8$ defined below only depend on line search parameters $\alpha$ and $\beta$,*

$$\delta_5 := \max \left\{ \frac{2 + (2/\delta_2)}{2\delta_2 - 17/16}, \ \frac{4\delta_3}{2\delta_2 - 17/16} \right\}, \qquad \delta_6 := \log \frac{1}{2\alpha(1-\beta)},$$

$$\delta_7 := 1 + \delta_4 \delta_6 + \delta_5, \qquad \delta_8 := 2 + 2\delta_4 \delta_6 + 2\delta_5 + \frac{2\delta_2 - 1/16 - \log \delta_2}{2\delta_2 - 17/16}. \tag{25}$$

Theorem 4.3 shows that the superlinear convergence rate of BFGS for a self-concordant function is of the form $(C/t)^t$ for some constant $C > 0$. Notice that from the definition of $t_0$ in (23), we know that $t_0 = \mathcal{O}(\Psi(\bar{B}_0) + (1 + D_0)^2 \log D_0)$. Hence, the superlinear convergence rate is of the order $\mathcal{O}((\frac{\Psi(\tilde{B}_0) + D_0(\Psi(\bar{B}_0) + (1+D_0)^2)}{t})^t)$, and we reach the superlinear convergence stage when $t \geq \Omega(\Psi(\tilde{B}_0) + D_0(\Psi(\bar{B}_0) + (1 + D_0)^2))$, which depends on the initial suboptimality $D_0$ and the initial Hessian approximation matrix $B_0$. To our knowledge, this is the first non-asymptotic global superlinear convergence rate of a quasi-Newton method without the assumption of strong convexity. Moreover, the superlinear rate in Theorem 4.3 is independent of the linear system chosen for the variables, and, hence, it is consistent with the affine invariance property of BFGS. Next, we present the superlinear convergence rate of BFGS for the special case of $B_0 = aI$, where $a > 0$.

**Corollary 4.4.** *Suppose Assumptions 2.2 holds, $\{x_t\}_{t\geq 0}$ are generated by BFGS with step size satisfying the Armijo-Wolfe conditions in (5) and (6), and $x_0 \in \mathbb{R}^d$ is an arbitrary initial point. If the*

*initial Hessian approximation matrix is $B_0 = aI$ where $a > 0$, the following result holds:*

$$\frac{f(x_t) - f(x_*)}{f(x_0) - f(x_*)} \leq \left( \frac{\delta_6 t_0 + \delta_7 \Delta_2 + \delta_8 D_0 (\Delta_1 + \frac{3(1+D_0)^2}{\alpha(1-\beta)})}{t} \right)^t ,$$

*where $t_0$ is defined in (23), $\{\delta_i\}_{i=5}^8$ are defined in (25) and $\Delta_1$, $\Delta_2$ are defined in (16), (20).*

## 5 Complexity Analysis

**Iteration Complexity.** Using Theorems 3.1, 3.3, and 4.3, we characterize the global iteration complexity of BFGS with inexact line search on self-concordant functions. These three results provide upper bounds, and the smallest of these bounds determines the complexity of BFGS. The smallest bound depends on the required accuracy relative to the problem and algorithm parameters. Specifically, for any initial point $x_0 \in \mathbb{R}^d$ and initial Hessian approximation matrix $B_0 \in \mathbb{S}_{++}^d$, to achieve a function value accuracy of $\epsilon > 0$, i.e., $f(x_T) - f(x_*) \leq \epsilon$, the number of iterations required, as per Theorem 3.1, is at most $T_1 = \mathcal{O}\left( \Psi(\bar{B}_0) + (1 + D_0)^2 \log \frac{1}{\epsilon} \right)$. The result in Theorem 4.3 eliminates the multiplicative factor in the $\log(1/\epsilon)$ term but requires a possibly larger additive constant, resulting in a complexity of $T_2 = \mathcal{O}(\Psi(\tilde{B}_0) + (\Psi(\bar{B}_0) + (1 + D_0)^2) D_0 + \log \frac{1}{\epsilon})$ Indeed, $T_2$ is smaller than $T_1$ when $\epsilon$ is small and $\log \frac{1}{\epsilon}$ becomes the dominant term. When $\epsilon$ is very small, the superlinear bound from Theorem 4.3 provides the best complexity, which is $T_3 = \mathcal{O}\left( (\log \frac{1}{\epsilon}) / \log \left( \frac{1}{2} + \sqrt{\frac{1}{4} + \frac{1}{\Psi(\bar{B}_0) + (\Psi(\bar{B}_0) + (1+D_0)^2) D_0} \log \frac{1}{\epsilon}}} \right) \right)$. Given these three bounds the overall iteration complexity of BFGS for the considered setting is $T = \min\{T_1, T_2, T_3\}$. Note that, for the special case of $B_0 = aI$ where $a > 0$ is an arbitrary positive constant, the complexity bounds denoted by $T_1, T_2, T_3$ can be further simplified as

$$T_1 = \mathcal{O}\left( \Delta_1 + (1 + D_0)^2 \log \frac{1}{\epsilon} \right), \; T_2 = \mathcal{O}\left( C_1 + \log \frac{1}{\epsilon} \right), \; T_3 = \mathcal{O}\left( \frac{\log \frac{1}{\epsilon}}{\log \left( \frac{1}{2} + \sqrt{\frac{1}{4} + \frac{1}{C_1} \log \frac{1}{\epsilon}} \right)} \right),$$

where $\Delta_1, \Delta_2$ are defined in (16), (20), and $C_1 := \Delta_2 + (\Delta_1 + (1 + D_0)^2) D_0$. For full iteration complexity details, see Appendix D.

**Line Search Complexity.** While the previous section characterized the complexity of BFGS under Assumption 2.2, analyzing its gradient complexity requires determining the number of gradient queries needed per iteration to obtain an admissible step size. In [42], the authors proposed an efficient log-bisection approach for step size selection in BFGS, satisfying the line search conditions in (5) and (6), and provided a complexity analysis. However, their results apply only to strongly convex functions with Lipschitz-continuous gradients and Hessians. In this section, we examine the line-search complexity of the log-bisection approach from [42] when the objective function is strictly convex and strongly self-concordant. Let $\Lambda_t$ denote the average number of iterations in Algorithm 1 required to terminate after $t$ iterations. The following proposition provides an upper bound for $\Lambda_t$.

**Proposition 5.1.** *Suppose Assumptions 2.2 holds. Let $\{x_t\}_{t \geq 0}$ be generated by BFGS with step size satisfying the Armijo-Wolfe conditions in (5) and (6) and is chosen by Algorithm 1. Let $\Lambda_t$ be the average number of the function value and gradient evaluations per iteration in Algorithm 1 after $t$ iterations. For any initial point $x_0 \in \mathbb{R}^d$ and initial Hessian approximation $B_0 \in \mathbb{S}_{++}^d$, we have that*

$$\Lambda_t = \mathcal{O}\left( 1 + \log \left( 1 + \frac{\Gamma}{t} \right) + \log \left( 1 + \log(1 + \frac{\Psi(\tilde{B}_0) + \Gamma}{t}) \right) \right),$$

*where $\Gamma = \mathcal{O}(D_0(\Psi(\bar{B}_0) + (1 + D_0)^2))$. As a corollary, for the special case of $B_0 = aI$ where $a > 0$, we have $\Lambda_t = \mathcal{O}(1 + \log(1 + \frac{\tilde{\Gamma}}{t}) + \log(1 + \log(1 + \frac{\Delta_2 + \tilde{\Gamma}}{t})))$, where $\tilde{\Gamma} = \mathcal{O}\left( D_0(\Delta_1 + (1 + D_0)^2) \right)$.*

This proposition implies the average number of iterations in Algorithm 1 is at most $\mathcal{O}(\log(1 + \Gamma))$, which is a constant depending on the initial suboptimality $D_0$ and the initial matrix $B_0$. Moreover, when the number of iterations $T$ exceeds $\Omega(\Psi(\tilde{B}_0) + \Gamma)$, the average number of function and gradient evaluations per iteration for Algorithm 1 is an absolute constant of $\mathcal{O}(1)$. Thus, even in the worst case, the gradient and iteration complexities remain of the same order, up to logarithmic factors.

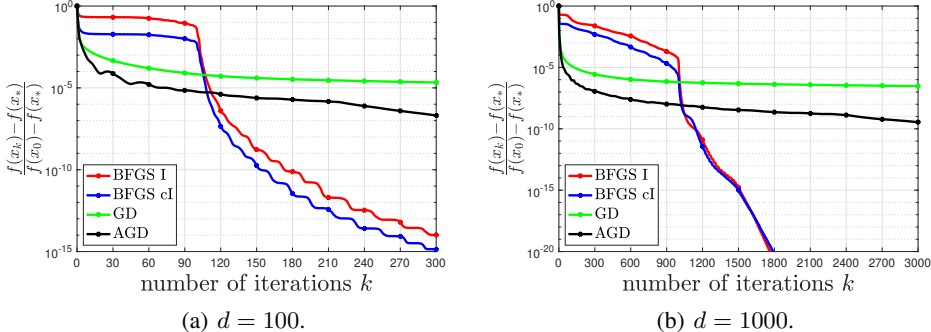

(a) $d = 100$.                (b) $d = 1000$.

Figure 1: Convergence rates of BFGS with different $B_0$, gradient descent and accelerated gradient descent for solving the hard cubic function with different dimensions.

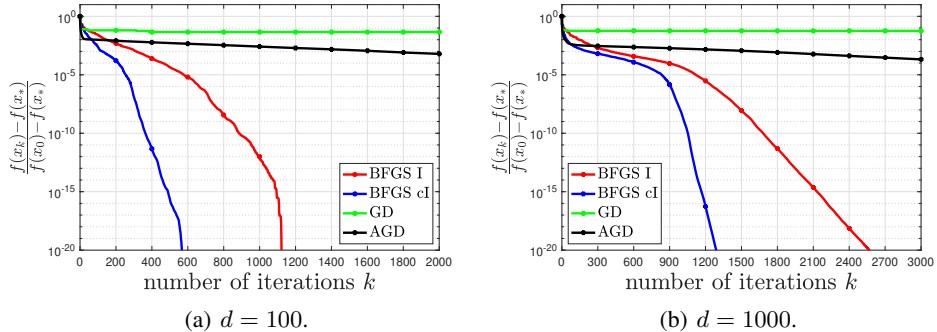

(a) $d = 100$.                (b) $d = 1000$.

Figure 2: Convergence rates of BFGS with different $B_0$, gradient descent and accelerated gradient descent for solving the logistic regression function with different dimensions.

## 6 Numerical Experiments

Next, we present numerical experiments applying BFGS to two functions satisfying Assumptions 2.2. We report our results using two different choices of initial Hessian approximation $B_0$: (i) $B_0 = I$, and (ii) $B_0 = cI$, where $c = \frac{s^\top y}{\|s\|^2}$, with $s = x_2 - x_1$, $y = \nabla f(x_2) - \nabla f(x_1)$, where $x_1, x_2$ are randomly selected. The line search parameters are also set as $\alpha = 0.1$ and $\beta = 0.9$. In our experiments, we also report the convergence paths of gradient descent (GD) and accelerated gradient descent (AGD), with step sizes determined using backtracking line search.

The first function that we study is the cubic function from [47]

$$f(x) = \frac{\omega_1}{12}\left[\sum_{i=1}^{d-1} g(v_i^\top x - v_{i+1}^\top x) - \omega_2 v_1^\top x\right], \quad \text{where } g(x) = \begin{cases} \frac{1}{3}|x|^3 & |x| \le \Delta, \\ \Delta x^2 - \Delta^2|x| + \frac{1}{3}\Delta^3 & |x| > \Delta. \end{cases}$$

Note that $g : \mathbb{R} \to \mathbb{R}$. We set the hypermeters of the objective function as $\omega_1 = 4, \omega_2 = 3, \Delta = 1$ and the vectors $\{v_i\}_{i=1}^n$ are set to be the orthogonal unit basis vectors of $\mathbb{R}^d$. We study this function as it serves as a benchmark for establishing lower bounds for second-order methods. The second loss is the logistic regression: $f(x) = \frac{1}{N}\sum_{i=1}^N \ln\left(1 + e^{-y_i z_i^\top x}\right)$, where $\{z_i\}_{i=1}^N$ are the data points and $\{y_i\}_{i=1}^N$ are their corresponding labels. We assume that $z_i \in \mathbb{R}^d$ generated with standard normal distribution and $y_i \in \{-1, 1\}$ generated with uniform distribution for all $1 \le i \le N$. We choose the number of data points as $N = d$. Note that both the hard cubic function and the logistic regression function are strictly convex and strongly self-concordant; see Appendix F.

The convergence paths for the cubic problem are shown in Figure 1 for various problem dimensions $d$. Initially, the performance of BFGS is worse than that of the first-order gradient descent and accelerated gradient descent methods. However, after approximately $d$ iterations, BFGS significantly outperforms the first-order methods. Notably, for this problem, the performance of BFGS with $B_0 = I$ and $B_0 = cI$ are nearly identical. Figure 2 shows the convergence paths for the logistic loss

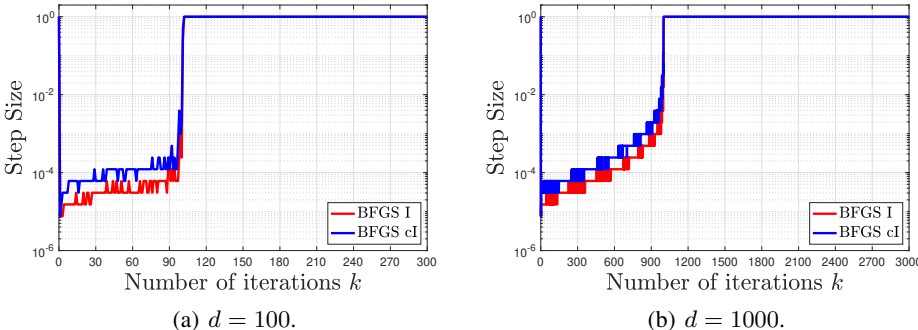

(a) $d = 100$.        (b) $d = 1000$.

Figure 3: Step size of BFGS with different $B_0$ using inexact line search for solving the hard cubic function with different dimensions.

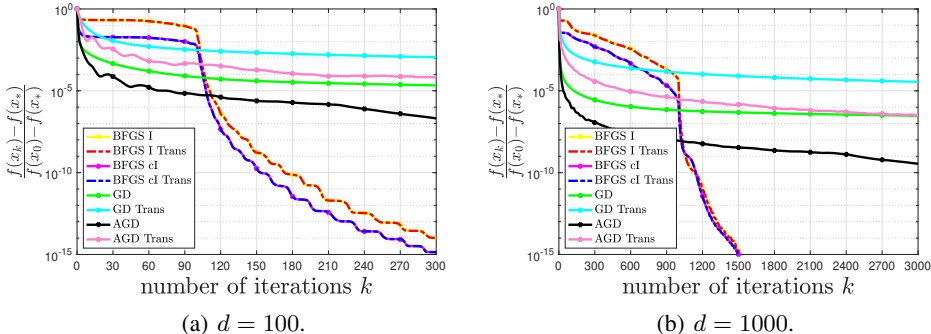

(a) $d = 100$.        (b) $d = 1000$.

Figure 4: Convergence rates of BFGS with different $B_0$, gradient descent and accelerated gradient descent for solving the hard cubic function with transformation matrix $A$.

across different problem dimensions $d$. Initially, BFGS performs similarly to first-order methods, but after several iterations, it outperforms them. Notably, in this experiment, BFGS with $B_0 = cI$ outperforms BFGS with $B_0 = I$. We also compared the performance of these different optimization methods with respect to the number of gradient evaluations and the time in seconds. Please check Figure 7 and Figure 8 in Appendix G and any other additional numerical experiments.

Moreover, we display the step sizes selected at each iteration by the inexact line search in the BFGS method in Figure 3. We observe that the step sizes are initially very small, then gradually increase, and after approximately $d$ iterations, they stabilize at 1 for nearly all subsequent iterations. This confirms our theoretical analysis: BEGS enters the superlinear convergence phase after about $d$ iterations, and there are only limited iterations where the unit step size didn't satisfy the weak Wolfe condition as proved in Lemma 4.2.

Finally, in Figure 4, we compare the performance of BFGS, GD, and AGD under a transformation matrix $A$ chosen to be a non-singular ill-conditioned matrix. We observe that the convergence trajectory of BFGS with this transformation is identical to that of the vanilla BFGS method, consistent with the affine invariance of quasi-Newton methods proved in Proposition 2.1. In contrast, the performance of GD and AGD degrades significantly under the transformation matrix, since first-order methods do not possess the affine-invariance property.

## 7 Conclusions

We established non-asymptotic global linear and superlinear convergence rates for the BFGS method on strictly convex and strongly self-concordant functions, using Wolfe step sizes. Our guarantees hold for any initial point $x_0 \in \mathbb{R}^d$ and any positive-definite initial Hessian approximation $B_0 \in \mathbb{S}_{++}^d$. Our analysis also respects the affine invariance of BFGS. A limitation is the reliance on strong self-concordance; extending results to standard self-concordance is a potential future direction.

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
