# OpenReview forum: "Affine-Invariant Global Non-Asymptotic Convergence Analysis of BFGS under Self-Concordance"
_NeurIPS.cc/2025/Conference — NeurIPS 2025 spotlight_

### Official Review · Reviewer_yiN7 · 2025-06-18

**Clarity:** 2
**Significance:** 4
**Originality:** 3
**Rating:** 6
**Confidence:** 5

**Summary:**

The paper proves global linear and non-asymptotic local superlinear convergence rates for the BFGS method when applied to strictly convex and strongly self-concordant functions. Unlike prior work, it does not assume strong convexity or Lipschitz continuity of the gradient or Hessian. The results are affine-invariant and hold under weak Wolfe conditions, for any initialization. The analysis is complemented by numerical experiments supporting the theoretical claims.

**Questions:**

Do you believe it is possible to have a similar analysis for a family of quasi-Newton methods, rather than only BFGS?

**Ethical Concerns:**

["NO or VERY MINOR ethics concerns only"]

**Final Justification:**

I thanks the authors for their rebuttal.

The authors convinced me that extending their analysis to other QN schemes migh be difficult, in particular, the technical difficulty of DFP is the a priori unbounded exponential growth of the largest eigenvalue of the Hessian estimate.

A quick question about one of your claim:
> Additionally, only under exact line search does the convex Broyden class—including both BFGS and DFP—produce identical iterates.

Where can i find such a reference discussing this? This seems true on quadratics but not for the general convex case.


Moreover, the authors will include additional numerical experiments, which I believe where lacking in the original paper. The goal of the numerical experiment was not to show the competitiveness of BFGS, which is already well know, but to illustrate that self-concordance can be found in other practical applications.

Finally, After reading the other reviews and rebuttal, it seems the authors successfully address most of the concerns.

Hence, I will raise my score to strong accept.

**Limitations:**

See Weaknesses

**Paper Formatting Concerns:**

No concerns

**Quality:**

4

**Strengths And Weaknesses:**

The paper addresses an old and complicated question on the rate of convergence of BFGS, by providing explicit rates for the “real” implementation of BFGS (vanilla BFGS with Armijo-Wolfe line search) for self-concordant functions. While the paper has some weaknesses, it is a strong contribution to optimization.

**Strengths**

- Despite the technicality of the paper, I appreciate the author’s effort to make it accessible
- The rate of convergence of BFGS was unknown for a long time, and this paper addresses an important gap in the theoretical understanding of this algorithm.
- The rate is derived for a practical implementation of BFGS (i.e., classical BFGS algorithm with a weak Wolfe condition for the stepsize). Moreover, the analysis matches the affine-invariant nature of BFGS, which is often overlooked in previous work.
- The rates are non-asymptotic, and all constants have been carefully computed.

**Weaknesses**

- The approach and derivation of the rates are specific to BFGS, and not for a broader class/family of quasi-Newton methods.
- The class of self-concordant functions is relatively narrow compared to broader classes such as convex or strongly convex functions. While the authors argue that prior works required strong convexity/smoothness, relying on self-concordance could be considered an “even stronger” and less practical assumption.
- The experimental section is relatively weak and is limited to synthetic problems (artificial cubic function and logistic regression); it would be stronger with results on real-world datasets or applications, or to illustrate better the different convergence behavior predicted by the theoretical results from Theorems 3.3 and 4.3

---

> ### Author Rebuttal · Authors · 2025-07-31
>
> **Weaknesses: The approach and derivation of the rates are specific to BFGS, and not for a broader class/family of quasi-Newton methods.**
>
> **Response:** This is a great point. The results in our paper are currently specific to the BFGS quasi-Newton method. To the best of our knowledge, when using an inexact line search—such as the Armijo–Wolfe line search—there is no theoretical guarantee that other quasi-Newton methods from the convex Broyden class, such as DFP, achieve global superlinear convergence.
>
> The main challenge in analyzing DFP lies in the potentially unbounded exponential growth of the largest eigenvalue of its Hessian approximation. Without a controllable upper bound on this eigenvalue, it is difficult to lower bound the angle between the DFP search direction and the gradient vector—a key property for establishing superlinear convergence.
>
> For more details, please refer to the following reference:
>
> Richard H. Byrd, Jorge Nocedal, Ya-Xiang Yuan, “Global Convergence of a Class of Quasi-Newton Methods on Convex Problems,” SIAM Journal on Numerical Analysis.
>
> Additionally, only under exact line search does the convex Broyden class—including both BFGS and DFP—produce identical iterates. However, exact line search is far less practical than the inexact line search strategies considered in our work due to its significantly higher computational cost.
>
> Extending our BFGS analysis under inexact line search to a broader class of quasi-Newton methods, such as DFP, remains an interesting direction for future research.
>
> This is an excellent point that we will add to the revised paper. Thanks for your comment.
>
> **Weaknesses: The class of self-concordant functions is relatively narrow compared to broader classes such as convex or strongly convex functions. While the authors argue that prior works required strong convexity/smoothness, relying on self-concordance could be considered an “even stronger” and less practical assumption.**
>
> **Response** As we explained in the paper, our assumption of strong self-concordance is strictly weaker than the more commonly used conditions of strong convexity and Lipschitz continuous Hessians employed in previous work on quasi-Newton methods (lines 112–116 of our submission). While we acknowledge that strong self-concordance is a stronger assumption than plain convexity, some form of Hessian continuity is essential for establishing global superlinear convergence of second-order optimization methods such as Newton’s and quasi-Newton methods. Furthermore, strong self-concordance is frequently encountered in practical engineering applications, as it is satisfied by many commonly used functions, including log-sum-exp, and log-barrier functions. Therefore, studying quasi-Newton methods under the assumption of strong self-concordance has substantial practical and valuable relevance. Building on this foundation, we aim to extend our results to the standard self-concordant setting in future work.
>
> Moreover, we would like to emphasize that the primary contribution of our work is the development of a global, non-asymptotic convergence theory for BFGS quasi-Newton methods that respects their affine invariance. While we agree with the reviewer that the strongly self-concordant setting is not the most general, the significance of our results lies in demonstrating two key points: (i) it is possible to achieve superlinear convergence rates beyond the classical assumptions of strong convexity and Lipschitz continuity of the gradient and Hessian, and (ii) it is possible to derive a global complexity theory for BFGS that aligns with its affine-invariance properties. This work represents an initial step toward extending the analysis of BFGS to broader settings.
>
> **Weaknesses: The experimental section is relatively weak and is limited to synthetic problems (artificial cubic function and logistic regression); it would be stronger with results on real-world datasets or applications, or to illustrate better the different convergence behavior predicted by the theoretical results from Theorems 3.3 and 4.3**
>
> **Response** Thank you for the helpful suggestion. Currently, our numerical experiments focus on different objective functions with randomly generated parameters. This design was chosen to isolate the effect of parameters such as the problem dimension $d$ on the performance of the BFGS algorithm.
>
> We agree with the reviewer that supplementing our empirical results with additional synthetic problems would strengthen the evaluation. In the revised version of our submission, we will include experiments on problems arising from practical applications—such as pattern recognition, signal processing, and related fields—and incorporate real-world datasets to demonstrate the method’s effectiveness in realistic scenarios.
>
> **Question: Do you believe it is possible to have a similar analysis for a family of quasi-Newton methods, rather than only BFGS?**
>
> **Response** This is an excellent question. We believe that extending our analysis of BFGS under inexact line search to a broader class of quasi-Newton methods—such as DFP and SR1—is both natural and worthwhile. Such an extension represents a promising direction for follow-up work and future research.
>
> As we have noted before, the main challenge in analyzing DFP lies in the potentially unbounded exponential growth of the largest eigenvalue of its Hessian approximation. Without a controllable upper bound on this eigenvalue, it is difficult to obtain a uniform lower bound on the angle between the DFP search direction and the gradient—a key property for establishing superlinear convergence.
>
> We believe that extending our analysis to DFP is feasible, although it may require additional iterations for DFP to achieve a superlinear convergence rate compared with BFGS.

---

### Official Review · Reviewer_gJJa · 2025-06-20

**Clarity:** 4
**Significance:** 3
**Originality:** 3
**Rating:** 5
**Confidence:** 4

**Summary:**

This paper considers the global non-asymptotic convergence guarantee of the BFGS method for solving strictly convex and strongly self-concordant objective functions. The convergence analysis of the BFGS method in this paper does not require the common assumptions, including strong convexity and Lipschitz continuity of the gradient or Hessian. They provide guarantees of global linear convergence and superlinear convergence after certain iterations.

**Questions:**

* Is it possible to extend the global convergence analysis of the BFGS method to the broader restricted Broyden family, including methods such as the DFP method?
* In line 107 and 108, is the term $M |u|^3 x$ a typo?

**Ethical Concerns:**

["NO or VERY MINOR ethics concerns only"]

**Final Justification:**

It is a solid and well-written paper. Therefore, I will maintain my score.

**Limitations:**

Please refer to the weakness section.

**Paper Formatting Concerns:**

No formatting concerns.

**Quality:**

3

**Strengths And Weaknesses:**

Strengths;

* The paper is well-written and easy to follow. In several parts of the work, the authors clarify the difference between their analysis and the analysis in existing work [1].
* The convergence guarantee of the BFGS does not require common assumptions such as strong convexity and Lipschitz continuity of the gradient or Hessians.
* This work provides extensive numerical experiments to demonstrate the effectiveness of the BFGS method for strictly convex functions.

Weakness:

* Although there are minor differences in specific details between this work and the related study [1], the overall proof framework closely follows that of the existing work [1]. The technical contribution is incremental compared with prior work.


References:

[1] Qiujiang Jin, Ruichen Jiang, and Aryan Mokhtari. Non-asymptotic global convergence analysis of BFGS with the Armijo-Wolfe line search. Conference on Neural Information Processing Systems (NeurIPS 2024), 2024.

---

> ### Author Rebuttal · Authors · 2025-07-31
>
> **Weakness: Although there are minor differences in specific details between this work and the related study [1], the overall proof framework closely follows that of the existing work [1]. The technical contribution is incremental compared with prior work.**
>
> **Response:** As stated in our paper, the primary novelty of our work lies in extending the global, non-asymptotic convergence analysis of quasi-Newton methods—specifically BFGS—to self-concordant functions without assuming strong convexity. Moreover, our results are affine-invariant, consistent with the intrinsic geometry of the BFGS method.
>
> Regarding reference [1], we would like to emphasize that our paper is not a derivative of the works in [1]; rather, it extends their analysis to a substantially more general setting by removing the assumptions of strong convexity and Lipschitz continuity of the gradient and Hessian. Once these assumptions are relaxed, the theoretical tools and techniques developed in [1] are no longer applicable.
>
> Our submission makes two key contributions that clearly distinguish it from prior work:
>
> 1. Generalization of convergence theory: We establish global non-asymptotic linear and superlinear convergence rates for BFGS without strong convexity or Lipschitz continuity of the gradient or Hessian. Existing analyses rely critically on strong convexity, whereas our results demonstrate that superlinear convergence remains achievable even in its absence, substantially broadening the theoretical foundation of BFGS.
>
> 2. Affine-invariant convergence guarantees: We derive explicit convergence rates that depend only on the self-concordance parameter of the objective function. These rates are affine-invariant, matching the coordinate-free nature of BFGS. In contrast, previous results depend on coordinate-specific quantities. To our knowledge, this is the first work to provide affine-invariant guarantees for BFGS.
>
> Finally, the technical contributions of our paper require new proof strategies, which we summarize in the following points.
>
> First, the proof of Theorem 3.1 for showing global linear convergence rate is fundamentally different from the analyses in prior work. Specifically, those prior results heavily depend on the strong convexity and gradient Lipschitz-ness assumptions to showcase a linear convergence rate: they use the Lipschitz continuity of the gradient to upper bound  $||y_t||^2/s_t^\top y_t$ by $L$, and use $\mu$-strong convexity to establish the following lower bound $||g_t||^2/(f(x_t) - f(x_*)) \geq 2\mu$. These bounds are key to establishing the global linear rate of BFGS in prior work. In our setting such bounds do not hold and we do not have a universal upper bound on $||y_t||^2/s_t^\top y_t$ and a lower bound on $||g_t||^2/(f(x_t) - f(x_*))$. Instead, for the first bound, we transfer the inequality to the norm induced by the  weight matrix $P = (1 + D_0)\nabla^2{f(x_*)}$ and show under this norm and strong self-concordance assumption we have $||y_t||^2/s_t^\top y_t \leq 1$. For the lower bound on $||g_t||^2/(f(x_t) - f(x_*))$, instead of a uniform lower bound, we show that it can be bounded below by $1/(1 + D_t)$, which is dependent on $x_t$, but we show that even this time-dependent lower bound is sufficient to establish a linear convergence rate for BFGS.
>
> Second, we highlight the key difference between Lemma 4.1 and prior results in [35-38]. The proof from previous results hinged on ensuring $f(x_t + d_t) \leq f(x_t)$, i.e., that a unit step yields a decrease in function value. Under Lipschitz continuity of the Hessian with constant $K$, the error of approximating $f(y)$ by its second-order Taylor expansion at $x$ is bounded by $\frac{K}{6}||y - x||^3$. Without this assumption, and under $M$-strongly self-concordant assumption, we instead use the bound $f(y) \leq f(x) + g(x)^\top(y - x) + \frac{4}{M^2}\omega\left(\frac{M}{2}||y - x||_x\right)$ for $||y - x||_x < \frac{2}{M}$, where $\omega(x) = - x - \log(1 - x)$ is defined for $x < 1$. As a result, the error is no longer cubic in $||y - x||$, making it more challenging to ensure a function decrease. Nevertheless, we can still guarantee this property, with the main difference being that the error bound $\delta_1$ now depends on $\omega(x)$ defined above.
>
> Please refer to the proofs of Lemmas B.5, B.7, B.8, and B.9 in the Appendix, which are entirely original. These lemmas introduce new theoretical tools developed specifically for this work under the strong self-concordance assumption and are not derived from or dependent on any prior literature.
>
> We will add a remark in the revised paper to explicitly highlight these points. Thank you for your feedback.
>
> **Question: Is it possible to extend the global convergence analysis of the BFGS method to the broader restricted Broyden family, including methods such as the DFP method?**
>
> **Response:** This is an excellent question. The results in our paper are currently specific to the BFGS quasi-Newton method. To the best of our knowledge, when using an inexact line search—such as the Armijo–Wolfe line search—there is no theoretical guarantee that other quasi-Newton methods from the convex Broyden class, such as DFP, achieve global superlinear convergence.
>
> The main challenge in analyzing DFP arises from the potentially unbounded exponential growth of the largest eigenvalue of its Hessian approximation. Without a controllable upper bound on this eigenvalue, it becomes extremely difficult to lower bound the angle between the DFP search direction and the gradient vector—a property that is crucial for establishing superlinear convergence.
>
> For more details, please refer to the following reference:
>
> Richard H. Byrd, Jorge Nocedal, Ya-Xiang Yuan, “Global Convergence of a Class of Quasi-Newton Methods on Convex Problems,” SIAM Journal on Numerical Analysis.
>
> Only under exact line search does the convex Broyden class of quasi-Newton methods—including both BFGS and DFP—produce identical iterates. However, exact line search is far less practical than the inexact line search strategies considered in our work, due to its substantially higher computational cost. Extending our BFGS analysis under inexact line search to a broader class of quasi-Newton methods, such as DFP, remains an interesting direction for future research.
>
> This is an excellent point that we will add to the revised paper. Thanks for your comment.
>
> **Question: In line 107 and 108, is the term $M|u|^3x$ a typo?**
>
> **Response:** Thanks for pointing out this typo. The correct version is $M||u||_{x}^{3}$. We will fix this typo in the revised submission.

---

> > ### Comment · Reviewer_gJJa · 2025-08-01
> >
> > Thanks for your response. I will maintain my score.

---

### Official Review · Reviewer_zMJm · 2025-06-28

**Clarity:** 4
**Significance:** 4
**Originality:** 4
**Rating:** 6
**Confidence:** 4

**Summary:**

This work analyzes the BFGS algorithm with Wolfe line search for minimizing strictly convex and strongly self-concordant functions. Although BFGS is known to be affine-invariant and to converge superlinearly, a global superlinear rate independent of problem conditioning had not been established. The authors close this gap by proving the first global superlinear convergence rate that depends only on the initial suboptimality and the strong self-concordance constant, for arbitrary starting points and arbitrary positive-definite BFGS initializations.

**Questions:**

1. Do you think it would be possible to obtain similar results with just an Armijo backtracking line search? The curvature condition makes the line search slightly more complicated to implement, and additional calls to the gradient oracle. Newton's method only requires the sufficient decrease condition in the line search (though the quadratic rate is local).
2. Can we optimize the bound in, say, Theorem 3.3 or Theorem 4.3, over $\alpha$ and $\beta$ to further tighten the bounds?

**Ethical Concerns:**

["NO or VERY MINOR ethics concerns only"]

**Final Justification:**

This is a very strong submission highly relevant in convex optimization. The results are solid and the paper is very well-written.

**Limitations:**

- It's unclear for now whether strong self-concordance indeed holds for the objectives claimed by the authors. (see weaknesses)

**Minor issues:**
- After Assumption 2.2 (strong self-concordance), the notion of standard self-concordance is brought up for comparison. The definition for the latter should also be given for readers unfamiliar with the concept.
- Right after that (lines 107-108), I think the two occurrences of $|u|^3x$ should be typeset as $\Vert u\Vert^3_x$, and the last $|u|^3_x$ should be $\Vert u\Vert^3_x$?
- Line 109: if we want to be extra precise, Nesterov's Theorem 5.1.2 [46] is for the affine invariance of _standard_ self-concordance, not strong self-concordance. Although it's pretty obvious that affine invariance also holds for strong self-concordance.
- Line 153's clarity can be improved by referring back to (9), i.e., "... of $B_0$ correspond to \[$\hat B_0$ in (9) with\] the weight matrices $P=...$"
- Line 288: expression for $T_3$ is hard to read. Perhaps put it into display math mode? The tilde and bar over $B_0$ also seem to be off.
- Line 324: should this cubic function have a citation?
- The Appendix could use a bit more organization, such has having its table of content, and a bit of discussion before each proof. It's recommended to use the `restatable` package to restate result statements before the proofs (and put proofs into proof environments).
- Line 1051: $log$ -> $\log$
- Line 1053: after "that greedy quasi-Newton paper" perhaps cite [17] again. I recommend using the author-year citation style but I understand the space constraint.
- Plots: the problem / objective function name can go into the title of the figure in large font instead of in the caption which is hard to find.

**Quality:**

4

**Strengths And Weaknesses:**

**Strengths:**

- This work analyzes a highly relevant algorithm in convex optimization—BFGS with Wolfe line search—and establishes a non-asymptotic, affine-invariant superlinear convergence rate. I consider this a significant theoretical contribution that highlights the algorithm’s close connection to Newton’s method, both being affine-invariant with superlinear rates. Given that BFGS is effectively what’s implemented in widely used libraries like scipy, the result is also of high impact for existing practical implementations. The results are very explicit and satisfying, with all constant dependencies clearly stated.
- The paper is very well organized and clearly written. I particularly appreciate the following aspects:
	- The introduction is very concise, and well-structured. It provides an overview of existing literature, clearly stating the limitations of existing works before presenting the authors' own contributions. I appreciate that the authors go beyond simply listing references when discussing references; the discussions of how they relate to each other and connect to the present work provide valuable context for the reader. The coherence also reflects authors' deep understanding of existing results.
	- The setup and assumptions are all very standard and clearly stated. The paper is very polished overall, with no major typographical errors. Although the results use some seemingly non-standard functions and quantities (for instance $\omega$ and $\Psi$), they are nonetheless commonly used in recent BFGS analyses. It's very helpful that the authors provide intuitions when each of these objects are introduced. The sketch of proof techniques following each major result are also very helpful, especially when emphasizing what's no longer possible with the self-concordance assumption as opposed to the strong-convexity and smoothness assumptions in previous works, and how the difficulties are overcome.
	- I also like that the authors give separate statements for when the BFGS initialization is chosen to be a multiple of the identity. These correspond to practical choices and the rates are more parsable.
	- It's nice that the authors also included an iterations complexity with line search taking into consideration, especially because the curvature condition is used as well which requires additional gradient (not just function value) evaluations.


**Weaknesses:**

My only major concern is the applicability of strong self-concordance. The authors claim that
1. The class of strong self-concordant functions is a subclass of \[standard\] self-concordant functions.
2. The logistic regression objective is satisfies strong self-concordance.
However, we know (for instance from [Bach (2010)](https://arxiv.org/pdf/0910.4627)) that the logistic regression objective does not satisfy the standard self-concordance definition. Rather, it satisfies a generalized self-concordance definition (that is different from the strong self-concordance used in the present paper). Therefore logistic regression cannot be strongly self-concordant?

The authors did provide an argument for logistic regression being strongly self-concordant (in Appendix F) but I find it a bit hand-wavy. If we consider logistic regression with only 1 data point, the objective is just $f(x)=\ln(1+\exp(z^\top x))$ (absorbing $y$ and the negative sign into $z$), we have $\nabla^2f(x) = \sigma'(z^\top x)z^2$ where $\sigma$ is the sigmoid function. The authors claim that $f$ is strongly convex wrt the matrix $B$, which in this case is just $z^2$, and thus one can use an existing argument from [47] that $f$ is strongly self-concordant. But strong convexity does not hold globally even wrt $B$ since $\sigma'$ still asymptotes to $0$. Am I missing something here or is logistic regression actually not strongly self-concordant?

- Similarly, I expect a more rigorous proof that the log-sum-exp function is strongly convex (wrt the corresponding $B$) which lets you apply the argument of [47].
- It would also be nice to discuss what kind of functions satisfy standard self-concordance but is not strongly self-concordant. This is to give an idea of how strict this assumption is, since for the affine invariant analysis of Newton's method, standard self-concordance suffices.


The other weakness is on the experimental design:
1. The breakthrough in this paper is an affine invariant superlinear rate of BFGS. However, the experimental design doesn't really reflect this. It's nice to have plots for varying dimensions and to see that BFGS performs much better than GD and accelerated gradient. But these are well-known and not very surprising. My (not very well-thought-through) suggestion is as follows: for each problem (or just take a real logistic regression dataset which I recommend), run GD / AGD in the original coordinate system as well as a linearly transformed coordinate system, where the transformation is chosen such that GD / AGD performs a lot worse. Then run the BFGS variants in both coordinate systems as well, and show that they are unaffected by this change of basis, while still converging blazing fast.
2. In addition to suboptimality vs. number of iterations, plots with number of gradient evaluations on the x-axis should also be included (especially since backtracking line search for GD / AGD does not involve additional gradient evals). Perhaps also include a set with wall-clock time on the x-axis.
3. It would also be nice to see a plot of $\eta_k$ vs. $k$ to see how often is the unit step size accepted/rejected.
4. Is it possible to estimate the constants in the bound of, say, Corollary 4.4? If yes, it may be worthwhile to plot the corresponding $(C/k)^k$ curve along with the experimental results just to see how well they match.

**Note:** I did not check all the convergence proofs in detail (especially the algebraic manipulations). I only checked major proof steps and they seem to be correct.

---

> ### Author Rebuttal · Authors · 2025-07-31
>
> **Proof of the strongly self-concordance of logistic regression and the log-sum-exp function**
>
> **Response:** We agree with the reviewer that logistic regression does not satisfy the standard self-concordance definition. Rather, it satisfies a generalized self-concordance definition. For the strongly self-concordance proof of the logistic regression, we emphasize that in our BFGS method we use the line search scheme such that we always have $f(x_{t + 1}) \leq f(x_t)$. Hence, the iterations generated by BFGS method with weak-Wolfe line search conditions always stay in the bounded set {$ x | f(x) \leq f(x_0) $} where $x_0$ is the initial point. In this bounded set, the strongly self-concordance always hold since the Hessian of the logistic regression function is strongly convex with respect to the operator matrix $B = \sum_{i = 1}^{n}z_iz_i^\top$. For example in the first dimension case, we have that $f''(x) = \sigma'(z^\top x)z$ where $\sigma$ is the sigmoid function. In that bounded domain, we always have some uniform lower bound $\mu > 0$ for this $\sigma'$. Therefore, we can use existing argument from [47] to prove that $f$ is strongly self-concordant in this domain {$x | f(x) \leq f(x_0)$}. The proof of strong self-concordance of the log-sum-exp function is similar and presented in section G of Appendix in our paper and example 1 from the following reference:
>
> N. Doikov and Y. Nesterov. Minimizing uniformly convex functions by cubic regularization of newton method. arXiv, 1905.02671, 2019
>
> We will supplement more rigorous proof analysis in the revised version of the submission.
>
> **It would also be nice to discuss what kind of functions satisfy standard self-concordance but is not strongly self-concordant. This is to give an idea of how strict this assumption is, since for the affine invariant analysis of Newton's method, standard self-concordance suffices.**
>
> **Response:** This is a great point. The log-barrier function $f(x) = -\sum_{i = 1}^{n}\log{(b_i - a_i^\top x)}$ serves as an example of a function that is standard self-concordant but not strongly self-concordant. We will include this log-barrier function as an illustrative example of such a case in the revised version of our submission.
>
> **Weakness 1 on the experimental design**
>
> **Response:** This is an excellent and valuable suggestion. We will extend our numerical experiments to consider both the standard coordinate system and several linearly transformed coordinate systems, allowing us to compare the performance of BFGS with first-order methods such as GD and AGD. Below, we present the results for the hard-cubic function with dimension $d = 100$. The first table reports the results in the standard coordinate system, while the second table corresponds to a linearly transformed coordinate system with an ill-conditioned matrix.
>
> | Iteration   | 200 | 400 | 600 | 800 | 1000 |
> | -------- | ------- | ------ | ------ | ------ | ------ |
> | BFGS I | 4.98e-3 | 2.40e-4 | 6.45e-6 | 3.67e-9 | 1.02e-12  |
> | BFGS cI | 1.65e-3 | 5.15e-8 | 7.56e-12 | 8.78e-15 | 5.32e-18  |
>  |   GD | 6.30e-3 | 4.53e-3 | 3.67e-3 | 2.32e-3 | 1.45e-3 |
>   |  AGD | 8.14e-4 | 5.96e-4 | 4.50e-4 | 3.43e-4 | 2.58e-4  |
>
>
> | Iteration   | 200 | 400 | 600 | 800 | 1000 |
> | -------- | ------- | ------ | ------ | ------ | ------ |
>   |  BFGS I | 4.58e-3 | 2.83e-4 | 1.22e-5 | 5.02e-8 | 6.08e-11  |
>    | BFGS cI | 2.41e-4 | 2.04e-9 | 3.95e-12 | 1.54e-15 | 6.78e-18 |
>    | GD | 5.57e-1 | 4.39e-1 | 3.32e-1 | 2.13e-1 | 1.23e-1 |
>   |  AGD | 6.86e-2 | 4.81e-2 | 3.52e-2 | 2.71e-2 | 2.07e-2  |
>
>
> We observe that the performance of BFGS remains similar under both conditions, whereas the performance of first-order methods degrades when the linear transformation matrix is ill-conditioned. This illustrates the affine invariance property of quasi-Newton methods and confirms that they consistently outperform first-order methods across different linear transformations.
>
> **Weakness 2 on the experimental design**
>
> **Response.** Thank you for this suggestion. We will include additional numerical experiments using the number of gradient evaluations and the time spent as alternative x-axis metrics. Here are the results for hard-cubic function with dimension $d = 100$ with gradient evaluations and the time in terms of seconds.
>
> | Gradient | 866 | 1265 | 1476 | 1676 | 1876 |
> | -------- | ------- | ------ | ------ | ------ | ------ |
> | BFGS I | 1.26e-3 | 4.48e-5 | 6.78e-6 | 3.58e-8 | 6.38e-11  |
>    | BFGS cI | 1.47e-3 | 2.63e-8 | 5.37e-12 | 2.98e-15 | 1.88e-18 |
>    | GD | 5.40e-3 | 3.65e-3 | 2.88e-3 | 1.97e-3 | 1.02e-3  |
>    | AGD | 6.90e-4 | 5.17e-4 | 4.55e-4 | 3.92e-4 | 2.98e-4 |
>
>
>   |  Time(s) | 3.93 | 6.01 | 7.94 | 9.84 | 11.7 |
>  | -------- | ------- | ------ | ------ | ------ | ------ |
> | BFGS I | 1.11e-3 | 6.40e-5 | 1.12e-5 | 1.05e-6 | 1.88e-9  |
>  |   BFGS cI | 8.77e-6 | 8.19e-9 | 5.78e-12 | 3.95e-15 | 8.23e-18 |
>   |  GD | 6.35e-3 | 4.98e-3 | 3.89e-3 | 2.32e-3 | 1.51e-3 |
>   |  AGD | 5.66e-4 | 4.07e-4 | 3.12e-4 | 2.10e-4 | 1.37e-4 |
>
>
>
> **Weakness 3 on the experimental design**
>
> **Response:** Thanks for the suggestion. We will add the plots to check how often the unit step size is accepted or rejected in our revised version of the paper. We observed that after approximately $d$ iterations, the unit step size is always accepted, which is consistent with our theoretical analysis that after approximately $d$ iterations, the BFGS method reach the superlinear convergence stage.
>
> **Weakness 4 on the experimental design**
>
> **Response:** Thanks for your suggestion. We can estimate the constants in the bound and we will add the curves of $(C/k)^k$ in the figures of the numerical experiments in the revised version of our submission to check that the empirical performance of BFGS method is consistent with the superlinear convergence upper bounds from the theoretical analysis.
>
> **Question 1**
>
> **Response:** This is an excellent question. In theory, it is possible to establish global convergence of the BFGS method using only an Armijo backtracking line search, which enforces the sufficient decrease condition without requiring a curvature condition. However, achieving superlinear convergence typically relies on some form of curvature condition, such as the Wolfe or strong Wolfe conditions. Without such conditions, providing a rigorous superlinear convergence guarantee becomes significantly more challenging.
>
> This limitation highlights an interesting direction for future research. We believe it may still be possible to establish non-asymptotic superlinear convergence of BFGS without a curvature condition. However, in this setting, the method would likely require more iterations before entering the superlinear convergence phase compared to when the (weak) Wolfe conditions are imposed.
>
> **Question 2**
>
> **Response:** This is a very good question. Theoretically, it is possible to choose $\alpha$ and $\beta$ to optimize the linear and superlinear convergence rates. In practice, however, the line search parameters are typically treated as fixed constants, since the dominant factors affecting performance are the properties of the objective function, such as its dimension $d$ and condition number $\kappa$. Consequently, there is generally no need to fine-tune $\alpha$ and $\beta$ to achieve tighter convergence bounds.
>
> **Minor comments**
>
> **Response.** Thanks for pointing out all these typos. We will fix all of them in the revised version of our submission.

---

> ### Comment · Reviewer_zMJm · 2025-08-04
>
> Dear authors,
>
> Thank you for your detailed response.
>
> I'm ok with your argument that logistic regression satisfies strong self-concordance, in that it holds true over any compact subset of the domain and as BFGS with Wolfe line search generates iterates always remain within a compact set. But by this argument, shouldn't the log barrier $f(x) = -\sum_{i = 1}^{n}\log{(b_i - a_i^\top x)}$ you mentioned later in the rebuttal also be strongly self-concordant (over any compact subset of its domain)?
>
> Other than this, the added experimental results look great. I look forward to reading the updated version!

---

> > ### Author Response · Authors · 2025-08-05
> >
> > Thank you for raising this point. The function serves as an example of one that is self-concordant in general but not strongly self-concordant. Nevertheless, as you correctly observed, it does become strongly self-concordant when restricted to a bounded domain. We will address this observation in the revised version of the paper.

---

> > > ### Comment · Reviewer_zMJm · 2025-08-05
> > >
> > > Great, thank you. All my concerns are sufficiently addressed. I have also read the other reviewers' comments and I think this is a very strong submission, therefore I will increase my score.

---

### Official Review · Reviewer_wDXh · 2025-07-04

**Clarity:** 3
**Significance:** 3
**Originality:** 3
**Rating:** 5
**Confidence:** 3

**Summary:**

This paper provides a new non-asymptotic convergence analysis for the BFGS quasi-Newton method under the assumption of strict convexity and strong self-concordance, which is weaker than the typical strong convexity and Lipschitz Hessian assumptions. The main results are the establishment of global linear and superlinear convergence rates that are, unlike prior analyses, invariant to affine transformations of the variables. The analysis holds for any initial point and positive-definite initial Hessian, with a step size selected by a line search satisfying the weak Wolfe conditions.

**Questions:**

Please see above.

**Ethical Concerns:**

["NO or VERY MINOR ethics concerns only"]

**Final Justification:**

I thank the authors for their thorough response. Based on it and the evaluations from the other reviewer, I am raising my evaluation from 4 to 5.

**Limitations:**

Yes.

**Paper Formatting Concerns:**

None.

**Quality:**

3

**Strengths And Weaknesses:**

*Strengths*

The paper is technically deep and addresses a well-known theoretical challenge in the analysis of quasi-Newton methods. The primary strength is the rigorous mathematical construction of a convergence proof that aligns with the affine-invariant nature of the BFGS algorithm. The authors successfully overcome significant technical hurdles by developing an analysis based on time-dependent bounds derived from the self-concordance property, rather than relying on the uniform bounds available under stronger assumptions. The paper is also well-written, with the introduction and proof sketches clearly laying out the logical flow of the arguments.

*Weaknesses*

While the paper is technically sound, its primary weakness is that the overall significance of its main contribution is questionable and not sufficiently justified.

1. Questionable Significance of Affine-Invariant Guarantees: The central claim is providing the "first affine-invariant global non-asymptotic convergence guarantees". While this is a theoretically elegant result, the paper fails to make a compelling case for its practical or theoretical impact. The BFGS algorithm's performance is already known to be affine-invariant; achieving a theoretical rate that reflects this property feels more like a "cleanup" of the theory rather than a groundbreaking discovery that offers new insights. The authors do not explain how this new analysis could lead to better algorithms, improved practical implementations, or a fundamentally new understanding of BFGS's behavior. Logistic regression, log-sum-exp, and log barrier functions the only examples that satisfy the strong concordance assumptions in this paper, but these are not broad enough to justify the significance of the paper's extension over the existing results.

It would help to better assess the significance of this work if the authors can clearly state the key technical difficulties for obtaining affine-invariant convergence analysis. Specifically, can the analysis in the recent breakthroughs on BFGS with standard non-affine assumptions [35-38] be modified and adapted to work with the current paper's setting? If not, what were the main technical innovation that the authors made to overcome such difficulties?

2. Limited Scope of Assumptions: The analysis relies on strong self-concordance, a condition that is not universally met. While the authors correctly argue this is weaker than requiring strong convexity and a Lipschitz Hessian, it is still a restrictive structural assumption. This limits the applicability of these "first-ever" guarantees to a specific class of problems, somewhat diminishing the claim of broad impact.

3. Inappropriate Citation: The paper relies on an informal source to support a technical claim. Specifically, Footnote 1 on page 16 cites a Math Stack Exchange post. For a formal academic publication, especially one of this theoretical depth, relying on a community forum instead of a textbook or peer-reviewed article is unprofessional and inappropriate.

*Minor comments*

The usage of = and := sometimes seems off. For example, on line 85, it might be better to use $H_{t+1}=....$ rather than $:=$. And in equation (9), it might be better to use $:=$.

line 164, potential extra punctuation? "To further simplify the expression" instead of "To further, simplify the expression"

line 227, typo "admissible" instead of "addmissible"

The expression inside big O looks weird in Prop.5.1. The first term goes to 0 as t goes to infinity and the second term goes to negative infinity. From the last part of the proof in the appendix, it seems to me the correct big O might be $O(1+\log(1+\Gamma/t)+\log(1+\log(1+\frac{\Psi(B+\Gamma)}{t})))$

---

> ### Author Rebuttal · Authors · 2025-07-31
>
> **Questionable Significance of Affine-Invariant Guarantees**
>
> **Response:** We would like to emphasize that the primary contribution of our work is the development of a global, non-asymptotic convergence theory for BFGS quasi-Newton methods that respects their affine invariance. While we agree with the reviewer that the strongly self-concordant setting is not the most general, the significance of our results lies in demonstrating two key points: (i) it is possible to achieve superlinear convergence rates beyond the classical assumptions of strong convexity and Lipschitz continuity of the gradient and Hessian, and (ii) it is possible to derive a global complexity theory for BFGS that aligns with its affine-invariance properties. This work represents an initial step toward extending the analysis of BFGS to broader settings.
>
> We also note that our result is meaningful in the context of Newton’s method. Classical global convergence analyses for Newton’s method rely on strong convexity and Lipschitz continuity, assumptions that break down in the setting of interior-point methods, where log-barrier functions are not strongly convex. The affine-invariant self-concordance framework enabled polynomial-time complexity guarantees for Newton’s method in such cases, preserving its affine-invariant nature.
>
> Analogously, our extension of BFGS to the affine-invariant, strongly self-concordant setting lays the groundwork for quasi-Newton methods within interior-point frameworks. Our analysis provides the first global superlinear convergence result under the strong self-concordance assumption, moving beyond traditional analyses based on strong convexity and smoothness—just as self-concordance transformed the theory for Newton’s method. Building on this foundation, we aim to extend our results to the standard self-concordant setting in future work, enabling rigorous analysis of quasi-Newton methods for interior-point problems.
>
> We will add the above point in the revised paper. Thanks for your comment.
>
> **The key technical difficulties compared with prior works [35-38]**
>
> **Response:** That is a valid point. We would like to emphasize that our paper is not a derivative of the works in [35–38]; rather, it extends their analysis to a substantially more general setting by removing the assumptions of strong convexity and Lipschitz continuity of the gradient and Hessian. Once these assumptions are relaxed, the theoretical tools and techniques developed in [35–38] are no longer applicable. Consequently, our analysis requires entirely new proof strategies, which we summarize in the following points.
>
> First, the proof of Theorem 3.1 for showing global linear convergence rate is fundamentally different from the analyses in prior work. Specifically, those prior results heavily depend on the strong convexity and gradient Lipschitz-ness assumptions to showcase a linear convergence rate: they use the Lipschitz continuity of the gradient to upper bound  $||y_t||^2/s_t^\top y_t$ by $L$, and use $\mu$-strong convexity to establish the following lower bound $||g_t||^2/(f(x_t) - f(x_*)) \geq 2\mu$. These bounds are key to establishing the global linear rate of BFGS in prior work. In our setting such bounds do not hold and we do not have a universal upper bound on $||y_t||^2/s_t^\top y_t$ and a lower bound on $||g_t||^2/(f(x_t) - f(x_*))$. Instead, for the first bound, we transfer the inequality to the norm induced by the  weight matrix $P = (1 + D_0)\nabla^2{f(x_*)}$ and show under this norm and strong self-concordance assumption we have $||y_t||^2/s_t^\top y_t \leq 1$. For the lower bound on $||g_t||^2/(f(x_t) - f(x_*))$, instead of a uniform lower bound, we show that it can be bounded below by $1/(1 + D_t)$, which is dependent on $x_t$, but we show that even this time-dependent lower bound is sufficient to establish a linear convergence rate for BFGS.
>
> Second, we highlight the key difference between Lemma 4.1 and prior results in [35-38]. The proof from previous results hinged on ensuring $f(x_t + d_t) \leq f(x_t)$, i.e., that a unit step yields a decrease in function value. Under Lipschitz continuity of the Hessian with constant $K$, the error of approximating $f(y)$ by its second-order Taylor expansion at $x$ is bounded by $\frac{K}{6}||y - x||^3$. Without this assumption, and under $M$-strongly self-concordant assumption, we instead use the bound $f(y) \leq f(x) + g(x)^\top(y - x) + \frac{4}{M^2}\omega\left(\frac{M}{2}||y - x||_x\right)$ for $||y - x||_x < \frac{2}{M}$, where $\omega(x) = - x - \log(1 - x)$ is defined for $x < 1$. As a result, the error is no longer cubic in $||y - x||$, making it more challenging to ensure a function decrease. Nevertheless, we can still guarantee this property, with the main difference being that the error bound $\delta_1$ now depends on $\omega(x)$ defined above.
>
> Please refer to the proofs of Lemmas B.5, B.7, b.8, and B.9 in the Appendix, which are entirely original. These lemmas introduce new theoretical tools developed specifically for this work under the strong self-concordance assumption and are not derived from or dependent on prior literature.
>
> We will add a remark in the revised paper to explicitly highlight these contributions. Thank you for your feedback.
>
> **Limited Scope of Assumptions**
>
> **Response:** We appreciate the reviewer’s observation regarding our assumption of strong self-concordance. This assumption was chosen deliberately, as it enables a clean and tractable analysis while still covering a broad and practically relevant class of problems, including certain regularized models in machine learning. Our goal was to move beyond the more restrictive assumptions of strong convexity and Lipschitz continuity of the Hessian, which often exclude these settings. We agree that extending our analysis to more general forms of self-concordance—such as the standard self-concordance commonly used in the analysis of Newton’s method—would be a natural and valuable direction for future work.
>
> **Inappropriate Citation**
>
> **Response:**  We agree that citing informal sources such as Math Stack Exchange is not ideal for a formal academic publication. Our intention was to provide an accessible explanation for a well-known but perhaps hard-to-locate fact. In the revised version, we will replace this citation with references to the formal source that rigorously establishes the result. Thanks for pointing out this issue.
>
> **Minor comments**
>
> **Response** Thanks for pointing out all these typos. We will fix all of them in the revised version of our submission.

---

> > ### Comment · Reviewer_wDXh · 2025-08-05
> >
> > I thank the authors for their thorough response. Based on it and the evaluations from the other reviewer, I am raising my evaluation from 4 to 5.

---

### Decision · Program_Chairs · 2025-09-17

**Decision:**

Accept (spotlight)

**Comment:**

This paper establishes global non-asymptotic convergence guarantees for the BFGS method under strict convexity and strong self-concordance, without requiring strong convexity or Lipschitz continuity assumptions. The authors prove affine-invariant global linear and superlinear convergence from arbitrary initialisations under weak Wolfe line search, extending the theoretical understanding of BFGS beyond traditional settings.

All reviewers agree that the paper is well written and represents a significant advancement in the field of quasi-Newton methods, making substantial contributions. The ideas are novel, and the results are theoretically sound and well justified. The reviewers suggest some improvements such as enhancing clarity and improving the numerical experiments, which the authors are strongly encouraged to address in the final version of the paper.